# Proteomic Analysis of Synovial Fibroblasts and Articular Chondrocytes Co-Cultures Reveals Valuable VIP-Modulated Inflammatory and Degradative Proteins in Osteoarthritis

**DOI:** 10.3390/ijms22126441

**Published:** 2021-06-16

**Authors:** Selene Pérez-García, Valentina Calamia, Tamara Hermida-Gómez, Irene Gutiérrez-Cañas, Mar Carrión, Raúl Villanueva-Romero, David Castro, Carmen Martínez, Yasmina Juarranz, Francisco J. Blanco, Rosa P. Gomariz

**Affiliations:** 1Department of Cell Biology, Faculty of Biology and Faculty of Medicine, Complutense University of Madrid (UCM), 28040 Madrid, Spain; selene@ucm.es (S.P.-G.); irgutier@ucm.es (I.G.-C.); macarrio@ucm.es (M.C.); ravillan@ucm.es (R.V.-R.); dcastr01@ucm.es (D.C.); cmmora@ucm.es (C.M.); yashina@ucm.es (Y.J.); 2Rheumatology Research Group, INIBIC-Biomedical Research Institute, CICA-University of A Coruña, 15006 A Coruña, Spain; valentina.calamia@sergas.es (V.C.); tamara.hermida.gomez@sergas.es (T.H.-G.); fblagar@sergas.es (F.J.B.)

**Keywords:** osteoarthritis, synovial fibroblasts, chondrocytes, VIP, CHI3L1, PTX3, complement system, decorin, cathepsin B, MMP2

## Abstract

Osteoarthritis (OA) is the most common musculoskeletal disorder causing a great disability and a reduction in the quality of life. In OA, articular chondrocytes (AC) and synovial fibroblasts (SF) release innate-derived immune mediators that initiate and perpetuate inflammation, inducing cartilage extracellular matrix (ECM) degradation. Given the lack of therapies for the treatment of OA, in this study, we explore biomarkers that enable the development of new therapeutical approaches. We analyze the set of secreted proteins in AC and SF co-cultures by stable isotope labeling with amino acids (SILAC). We describe, for the first time, 115 proteins detected in SF-AC co-cultures stimulated by fibronectin fragments (Fn-fs). We also study the role of the vasoactive intestinal peptide (VIP) in this secretome, providing new proteins involved in the main events of OA, confirmed by ELISA and multiplex analyses. VIP decreases proteins involved in the inflammatory process (CHI3L1, PTX3), complement activation (C1r, C3), and cartilage ECM degradation (DCN, CTSB and MMP2), key events in the initiation and progression of OA. Our results support the anti-inflammatory and anti-catabolic properties of VIP in rheumatic diseases and provide potential new targets for OA treatment.

## 1. Introduction

Osteoarthritis (OA) is the most common musculoskeletal disorder that affects nearly 300 million people in the world population that are over 60 years of age. It causes great disability and a substantial reduction in the quality of life, representing a global public health problem with no current treatment. It is classically established that chronic joint overload alters its mechanical function, triggering joint inflammation and cartilage degeneration [1]. Current information proposes that OA is a more complex pathology that includes other causal factors, such as genetic and metabolic factors. The confluence of all these factors generates the disruption of the homeostasis of the whole joint, including the cartilage, subchondral bone, synovium, meniscus and the infrapatellar fat pad, by the release of catabolic and inflammatory factors [2,3].

Although the final consequence is the loss of articular cartilage, all joint tissues, including bone and synovium, participate in the production of these factors concerning both different cellular types and the extracellular matrix (ECM) [4,5,6]. In OA, articular chondrocytes (AC) and synovial fibroblasts (SF) mainly release innate-derived immune mediators that initiate and perpetuate inflammation, inducing cartilage ECM degradation [5,7,8,9]. The fact that SF are active drivers of joint destruction in rheumatoid arthritis (RA) is well established [10], but their behavior in OA patients is less understood.

The ECM is a complex mixture of macromolecules secreted by cells into the extracellular space that provides the supportive architecture on which cells adhere, migrate, and regulate tissue development [11], interacting with cell surface receptors on joint resident cells. Therefore, the ECM not only functions as a structural support for a group of cells in a tissue, but also actively communicates with the cells to ensure homeostasis. Thus, variations in the composition and physical properties of the ECM lead to the development of many diseases, including cancer and rheumatic diseases, such as OA and RA, among others [12,13,14]. In this sense, the integrity of ECM composition is crucial to the regular function of load-bearing tissues, such as cartilage in OA [15].

The glycoprotein fibronectin (Fn) is a component of the pericellular matrix that plays an important role in the maintenance of the mechanical properties of the cartilage [16]. During OA progression, the injury induces tissue proteolysis and ECM damage, generating ECM-degradation products, including Fn fragments (Fn-fs), that induce cytokine and proteinase expressions, chronifying the inflammation [12,17,18].

Joint tissues are sources of bioactive neuropeptides, such as neuropeptide Y, pituitary adenylate cyclase-activating polypeptide and vasoactive intestinal peptide (VIP) that induces changes in the cell metabolism in degenerative conditions such as OA [19]. VIP and its G protein-coupled receptors (GPCRs), VPAC1, and VPAC2, form a signaling axis that modulates both the innate and acquired immunity in several inflammatory/autoimmune diseases, including OA [20,21]. VIP sources in the joint comprise both nerve fibers of the sympathetic nervous system and a cellular origin, including lymphocytes and SF from OA and RA patients [20,21,22]. Functionally, a decrease in the number of these nerve endings is described in OA and RA [19]. Moreover, VIP levels are reduced in cartilage and synovial fluid of OA patients, compared to healthy controls, which could participate in the pathology development [23]. Clinically, VIP levels in synovial fluid and cartilage of OA patients are negatively related with joint impairment, being a prospective indicator of disease severity [24].

Given the absence of specific therapies for the treatment of OA, the exploration of biomarkers that enable the development of new therapeutical approaches is crucial. In this context, we proposed to analyze the set of secreted proteins in chondrocyte and synoviocyte co-cultures, joint cell types involved in crucial physiological processes in OA. Moreover, we also studied the modulating role of the endogenous peptide VIP in this secretome, providing novel VIP-modulated proteins involved in the main events that take place during the OA pathology: the inflammatory process, the activation of the complement system, and the cartilage ECM-degradation.

## 2. Results

### 2.1. SF-AC Co-Cultures Secretome Profiling

SF and AC grown in a stable isotope labeling with amino acids (SILAC) medium were put together in co-culture and treated with 45 kDa Fn-fs as a stimulus of inflammation and ECM destruction in the presence and absence of VIP to elucidate which proteins are modulated by this neuropeptide. A total of 115 proteins were detected in the secretome of the co-cultures (Table 1, Figure 1).

### 2.2. Modulation of SF-AC Co-Cultures Secretome by VIP

Among the proteins detected by SILAC, 28 proteins were significantly modulated by VIP in one of the replicates (forward or reverse) for the four patients (Appendix A). Nine of these proteins were consistently modulated in both replicates (forward and reverse) decreasing 8 of them, as shown in the ratio Fn-fs+VIP:Fn-fs (Figure 2a, Table 2, Appendix A). Proteins downregulated by VIP included those involved in the immune response: chitinase-3-like protein 1 (CHI3L1), pentraxin-related protein 3 (PTX3), complement C1r subcomponent (C1r), complement C3 (C3), sulfhydryl oxidase 1 (QSOX1), and cathepsin B (CTSB). Among them, CHI3L1, PTX3 and C3 were also implicated in inflammation, and C1r, C3 took part in complement activation. In addition, VIP also decreased proteins involved in ECM degradation, including DCN, CTSB, and MMP2 (Figure 2b).

### 2.3. VIP-Modulated Inflammatory Proteins

Among the proteins modulated by VIP in the SILAC analysis, we decided to validate those involved in the inflammatory response, CHI3L1 and PTX3, which is a common process that takes place in OA, worsening the disease progression. Firstly, we evaluated the constitutive production of these proteins by SF and AC cultured alone, as well as in co-culture for 48 h, by ELISA and Multiplex for CHI3L1 and PTX3, respectively. The results showed that AC produced more CHI3L1 than SF. This increase was also observed in the co-culture (Figure 3a). No differences were observed in PTX3 (Figure 3b). Next, the SF-AC co-cultures were treated with 45 kDa Fn-fs, as a pro-inflammatory stimulus, in the presence and absence of VIP. No effects were observed with the Fn-fs treatment alone, while in presence of VIP, the production of both CHI3L1 and PTX3 was decreased in the secretome of the co-cultures, corroborating the anti-inflammatory effect of VIP (Figure 3c,d). No significant effects were observed when the SF and AC were cultured alone (Appendix A).

### 2.4. VIP-Modulated Complement System Proteins

In addition to the inflammatory proteins mentioned above, results derived from the SILAC analysis showed that VIP tends to decrease the complement cascade proteins CFB, complement C1r, C1s subcomponents, C3, as well as the complement-related protein clusterin. Among them, VIP significantly decreased C1r, C1s, C3 and clusterin in any of the replicates for the 4 patients, as observed in the ratio Fn-fs+VIP:Fn-fs, and the complement system proteins C1r and C3 in both replicates (Appendix A). Regarding the constitutive production, no significant differences were observed between the different cultures in the secreted levels of C1r nor in C3 (Figure 4a,b). On the other hand, the results obtained from ELISA showed a decrease in the production of C1r in the SF-AC co-cultures treated with Fn-fs+VIP in comparison to the inflammatory stimulus alone, while no effects were observed with the Fn-fs (Figure 4c). However, despite VIP tended to decrease the levels of C3, no significant differences were observed by Multiplex analysis (Figure 4d). No significant effects were observed in the isolated cultures either (Appendix A). 

### 2.5. VIP-Modulated ECM Degradation Proteins

Degradation of cartilage ECM is one of the key events that take place in the OA pathology. SILAC analysis showed that VIP modulated some of these proteins, significantly decreasing DCN, MMP2 and CTSB in the secretome from SF-AC co-cultures. Therefore, these molecules were validated by ELISA for DCN and CSTB, and Multiplex for MMP2. Firstly, we evaluated the constitutive production of these proteins by the SF and AC cultured alone as well as in co-culture for 48 h. We observed that SF and AC had an additive effect in the production of DCN (Figure 5a), while the SF produced more CTSB and MMP2 than the AC, whose production was also reduced in the co-culture (Figure 5b,c). Next, the SF-AC co-cultures were treated with 45 kDa Fn-fs as a stimulus of ECM destruction in the presence or absence of VIP. While no effects were observed with the Fn-fs treatment alone, the presence of VIP reduced the production of DCN, CTSB and MMP2 in the secretome of the co-cultures, corroborating the protective effect of VIP in the ECM degradation (Figure 5d–f). A decrease in the CTSB and MMP2 production was also observed in the secretome of SF cultured alone in the presence of VIP when stimulated with Fn-fs (Appendix A), while this decrease was not significant in the AC isolated cultures (Appendix A). No significant effects were observed in the levels of DCN in the case of the isolated cultures (Appendix A). 

## 3. Discussion

As far as we know, this is the first study that analyzed the proteins detected in the secretome of SF-AC co-cultures in OA patients. We described a total of 115 proteins, 74 of which had been previously reported in the secretome of IL-1β-stimulated AC from healthy donors by SILAC [25]. When comparing with our results, 39 were commonly identified in both secretomes, while 76 proteins were exclusively detected in the present Fn-fs-stimulated SF-AC co-cultures, and 35 in the previous IL-1β-stimulated AC cultures (Appendix A). 

Discrepancies regarding the number of proteins identified could be explained by the different experimental designs used in the studies. The main differences include the stimuli (IL1β in the previous analysis vs. Fn-fs in the present one), the cell type (AC vs. SF-AC co-cultures), as well as the samples origin (healthy donors vs. OA patients). In another proteomic study, 30 proteins were over-expressed in the lysate of SF from RA patients compared with controls [26].

In the present study, we also reported, for the first time, a proteomic analysis by SILAC of the effect of a neuropeptide present in the joint in OA patients. Among the proteins detected in the SF-AC co-cultures secretome, VIP consistently decreased 8 of them, including proteins involved in the inflammatory process and complement activation, as well as in the cartilage ECM degradation, key events in the initiation and progression of OA. 

Regarding pro-inflammatory proteins, we described the modulation by VIP of two immune-related proteins: CHI3L1 (also known as human cartilage glycoprotein-39 or YKL-40) and PTX3.

CHI3L1 is produced by several cell types, including synoviocytes, chondrocytes, osteoblasts, macrophages, monocytes, neutrophils, and vascular smooth muscle cells [27,28,29,30]. Increased levels of CHI3L1 were reported in several pathologies, including rheumatic diseases and cancer, being involved in inflammation, tissue remodeling and tissue injury in health, as well as during disease [29,31,32,33]. CHI3L1 is also related to bone resorption activity [32]. Its levels in serum and synovial fluid correlate with OA and RA severity [34,35,36,37,38,39,40,41]. Moreover, higher levels of CHI3L1 were addressed in OA synovial fluid and sclerotic osteoblasts compared to healthy donors [27,40]. In addition, CHI3L1 is positively associated with inflammation and cartilage destruction mediators in OA [42]. All in all, CHI3L1 represents a potential biomarker in several pathologies, including rheumatic diseases, such as psoriasis, psoriatic arthritis, RA, and OA [43,44]. 

Conversely, PTX3 belongs to the group of acute-phase proteins pentraxins, which play a key role in inflammation, also being involved in cell proliferation, migration, and ECM remodeling [45,46]. Production of PTX3 is induced in different cell types in response to pro-inflammatory mediators, including macrophages, endothelial cells and synoviocytes. Increased levels of PTX3 were reported in patients with systemic inflammation, including septic shock, myocardial infarction, and systemic vasculitis [47]. Regarding rheumatic diseases, PTX3 is involved in osteoblast proliferation in osteoporosis [46] and its expression is induced by TNF-α in OA SF [48]. In RA, increased levels of PTX3 are associated with disease severity [45,47,48].

In the present study, we corroborated the production of CHI3L1 and PTX3 by both OA-SF and -AC, where AC produces more CHI3L1 than SF, consistent with previous studies, suggesting that cartilage is the main source of this protein [42]. In addition, its levels in AC are related to tissue degeneration in OA [49]. We validated the SILAC results by ELISA or Multiplex, confirming that VIP decreases the pro-inflammatory mediators CHI3L1 and PTX3 in the secretome from SF-AC co-cultures, validating the anti-inflammatory properties of VIP [20,21,50,51,52]. 

Complement system proteins are involved in the pathogenesis of OA by means of ECM degradation, AC and SF inflammatory responses and synovitis, cartilage angiogenesis, bone remodeling and osteophyte formation, cell lysis, and stem cell recruitment. Several studies have focused on the complement pathway as a target for the treatment of rheumatic diseases [53]. During complement activation, its inflammatory cleavage products bind to their receptors and mediate several inflammatory effects. The complement can be activated by cartilage ECM components and their cleavage products [54,55,56,57,58]. Specifically, fibronectin binds to the C1q complement subcomponent [59,60]. Cartilage, synovium and bone are key sources of the classical and alternative complement pathway components [54,61], produced by articular cells, including AC and SF in healthy donors, RA, and OA patients [62,63,64,65]. Increased levels of complement factors have been reported in the synovial fluid from OA and RA joints and acute knee injuries [54,66]. In addition, complement activation is also associated with inflammation and osteochondral fractures in OA [67], and antiangiogenic, anti-inflammatory, and anti-catabolic effects of chondroitin sulphate have been related to the reduction of complement components, including CFB, C1r, C1s and C3 [25].

C1 is the first component of the classical pathway of the complement system, a multimolecular protease consisting of two catalytic subunits, C1r and C1s, and recognition protein, C1q [68]. In a previous proteomic study complement C1r was the most significantly upregulated protein in OA synovial fluid, whose levels correlated with severity [69]. Increased levels of C1 complex were also previously described in serum and synovial fluid from RA patients [70,71]. 

On the other hand, cleavage of C3 is the common point of the three complement activation pathways (classical, alternative, and lectin pathway) [54]. C3 is one the most expressed complement components in OA osteochondral biopsies, cartilage, AC and SF [61], and may be involved in bone remodeling in OA [72]. Moreover, in OA and other arthritic diseases, C3 cleavage fragments in serum and synovial fluid are related to disease severity, pain, and clinical symptoms [73,74,75,76,77,78,79].

Consistent with previous findings, our results derived from SILAC analysis in the SF-AC co-cultures corroborated the production of the complement cascade proteins CFB and CFH, complement C1r and C1s subcomponents, C3, and clusterin. However, no presence of C2, C4, and C1q was detected, as reported by other authors in human AC [25,64]. Regarding VIP, it tends to reduce these complement cascade proteins, consistently decreasing complement C1r, as well as C3, results corroborated by ELISA for C1r. C3 Multiplex analysis also showed a decrease, although it was not significant, which was due to the limited number of samples. These data demonstrate, for the first time, the regulatory character of VIP in the complement system, an important component of innate immunity.

Finally, we reported the VIP-mediated reduction of three proteins involved in ECM homeostasis: DCN, CTSB and MMP2. 

ECM-degradation is a key feature in OA and an important focus of investigation due to the irreversible nature of the damaged cartilage. DCN is a structural protein highly present in the articular cartilage interterritorial and pericellular ECM [80,81,82,83,84,85,86,87]. This small, leucine-rich proteoglycan is involved in the regulation of different biological functions, including ECM organization, cell adhesion, migration and proliferation [87]. In rheumatic diseases, it was detected in the synovial fluid from OA and RA patients [88]. Increased DCN expression was described in OA cartilage [89,90,91], as well as in fibroblast-like chondrocytes, being upregulated in late stages of the disease [87]. In addition, DCN possess pro-inflammatory properties through activation of TLR4 and the NF-κB pathway [88]. Previous studies have reported a VIP-mediated downregulation of TLR4 expression inhibiting NF-κB signaling [92]. Increased levels of DCN-specific antibodies have been identified in the synovial fluid from patients with inflammatory rheumatic diseases [93]. Soluble isoforms of DCN released by ADAMTSs and MMPs in damaged cartilage can act as endogenous danger signals in OA and RA [88,89,90,93,94]. Binding of DCN to C1q has also been reported, but without induction of complement activation [58]. Conversely, addition of DCN to synovial fluid inhibits SF expansion in OA patients [95]. 

Cysteine cathepsins and MMPs are highly involved in the development of OA [18,96,97,98,99,100,101]. CTSB is the main cysteine peptidase in OA cartilage [102], involved in cartilage-ECM degradation including collagen and aggrecan [103,104]. In turn, fragments derived from collagen type II induce expression of cathepsins, including CTSB, in OA AC [99]. In addition, CTSB degrades the MMPs inhibitors TIMP-1 and TIMP-2, promoting angiogenesis, mineralization and osteophyte formation [105]. However, it is considered a perpetuator rather than an initiator of cartilage destruction [102,106]. CTSB is reported to be a marker of chondrocyte differentiation since its levels increase by subculturing cells in monolayers, suggesting dedifferentiation toward a SF phenotype [106,107]. In addition, CTSB action is associated with joint inflammation [108], and elevated levels of cysteine cathepsins are present in synovial fluid and SF from RA patients [100,109,110,111,112] as well as in the synovial membrane from RA and OA patients compared to controls [113,114]. Moreover, increased CTSB activity and expression are shown in AC and cartilage from OA patients [102,115,116,117,118,119,120], and its levels in serum and synovial fluid correlate with severity and joint inflammation [119]. Furthermore, it is hypothesized that anomalous trafficking of cathepsin B found in pathologies, such as OA, depend on alternative splicing of its pre-mRNA [121]. 

MMPs are the main proteases involved in the degradation of collagen type II, produced by chondrocytes as well as by synovial cells, whose contribution in rheumatic diseases has been widely studied [122,123]. MMP2, also known as gelatinase A, degrades denatured collagens and gelatins, as wells as other ECM components [122,124]. Several studies have noted increased MMP2 levels in cartilage [98,125,126,127,128], serum [129], synovial fluid [130,131,132,133], as well as in the synovium and pannus-like tissue in OA [128,134]. This increased expression was also detected in SF from RA and OA patients [135,136], in synoviocytes co-cultured with fibroblasts [137] and platelets [130], as well as in AC co-cultured with subchondral bone osteoblasts in OA [138]. Consequently, MMP2 is defined as a potential biomarker for OA [139,140,141], and is positively correlated to histopathology severity in OA cartilage [142]. Conversely, Thorson et al. detected lower MMP2 plasma levels in OA patients compared to controls, suggesting a secondary role of this MMP [143].

All in all, regarding proteins involved in ECM homeostasis, we showed that SF and AC have an additive effect in the production of DCN in co-culture. We also demonstrated that SF produce more MMP2, which is inhibited by the presence of AC in the co-culture, compared to SF cultured alone. Regarding VIP, it also reduces the levels of DCN, CTSB and MMP2 in the SF-AC co-cultures. Decrease in MMP2 expression mediated by VIP was also reported in renal-cell carcinoma [144]. The present study supports previous data from our laboratory reporting the beneficial effects of VIP in the protection of several components of the ECM, modulating uPA, MMP9 and MMP13 as well as ADAMTS-4, -5, -7, and -12 production. [8,145]. Moreover, since DCN, CTSB and MMP2 also promote the inflammatory process in OA, reduction mediated by VIP again corroborates its well-known anti-inflammatory role in rheumatic diseases. 

We used 45 KDa Fn-fs as a pro-inflammatory and degradative stimulus as previously described [18]. However, despite the fact that no significant effects were detected in the proteins analyzed in this study with the Fn-fs treatment alone, this stimulus is necessary for VIP to exert its effects (data not shown). 

Since a pioneering study demonstrated the beneficial role of VIP in a murine arthritis model, downregulating both the inflammatory and the autoimmune components of the disease [146], different studies valued its therapeutic potential. Gradually these results were corroborated in different inflammatory/autoimmune pathologies. In rheumatic diseases, its protective effect was also reported in bone erosion through NF-κB and AP-1 modulation [147]. In early arthritis (EA) patients, disease activity inversely correlated with VIP serum levels and after a two-year follow-up, those patients with low baseline levels of VIP showed higher disease activity and received more intensive treatment [148]. Confirming these data, an association between serum VIP levels and variants in the VIP gene was reported [149]. 

Although most of the effects point to a beneficial role of the VIP axis specifically in OA, it was described in murine models that VIP might promote mechanosensitivity and pain [150]. In patients, it was also reported a pro-inflammatory role of VIP [151]. However, VIP is downregulated in the synovial fluid of patients, which entails a rise in the expression of pro-inflammatory cytokines contributing to OA development [24]. Thus, it can be hypothesized that a defeat of VIP contributes to the pathogenesis and that the reestablishment of VIP levels could pause or suppress disease evolution [23,24]. In addition, since no significant effects of the VIP treatment were observed in the isolated SF and AC cultures, with the exception of CTSB and MMP2 in Fn-fs+VIP-treated SF, our study highlights the significance of both cell types in the microenvironment of the OA joint, further supporting the worth of our study in co-cultures.

Some limitations of the study should be noted, including the limited number of samples, the lack of controls, and the low number of identified proteins. Undoubtedly, further studies employing high resolution equipment would allow the identification of a higher number of proteins in the secretomes of the co-cultures not detected in the present study. In addition, future studies with more samples and comparisons between OA patients and healthy donors would be also of interest. Nevertheless, we honestly believe that the limitations due to a rapidly obsoleting technology do not reduce the interest and impact of our findings.

In summary, the novelty of the current study consists in the identification of the proteins released by two cell populations present in the OA joint and key in the pathology of the disease. Our results support the anti-inflammatory and anti-catabolic properties of VIP by reducing several crucial mediators involved in inflammatory/immune response and ECM degradation. Finally, we demonstrate, for the first time, the modulation of the complement cascade proteins mediated VIP.

## 4. Materials and Methods 

### 4.1. Subjects and Samples Procurement

Osteoarthritic cartilage and synovial tissue were obtained from the knee of ten patients (five women and five men) aged between 50 and 92, undergoing leg amputations caused by trauma or individuals who died and were tissue and organ donors. All tissue samples were provided by the Autopsy Service at Hospital Universitario de A Coruña. Informed consent was obtained from the patients before surgery. OA patients were diagnosed following the American College of Rheumatology (ACR) criteria for OA classification [152]. The study was approved by the local Ethics Committee (Galicia, Spain).

### 4.2. Cell Cultures

AC were isolated as previously described [153]. Briefly, cartilage surfaces were rinsed with a saline buffer; scalpels were used to cut parallel vertical sections 5 mm apart from the cartilage surface to the subchondral bone. These cartilage strips were dissected from the bone, and the tissue was incubated with trypsin at 37 °C for 10 min and then digested with type IV clostridial collagenase. The release of AC from cartilage was achieved after 16 h of digestion in an incubator at 37 °C on a microplate shaker. SF cultures were established by explant growth of synovial biopsies, cultured in Dulbecco’s modified Eagle’s medium (DMEM) with 25 mM HEPES and 4.5 g/L glucose, completed with 10% heat-inactivated fetal bovine serum (FBS) (Lonza Ibérica SAU, Barcelona, Spain), 1% L-glutamine and 1% penicillin-streptomycin (Invitrogen, Carlsbad, CA, USA), at 37 °C and 5% CO_2_. After three passages, residual contamination by macrophages was avoided, previously assessed by flow cytometry analysis of SF with a purity of 95% [92]. 

### 4.3. Chondrocyte and Synovial Fibroblasts Stable Isotope Labeling by Amino Acids in Cell Cultures (SILAC)

AC and SF were labeled as previously described [154]. Briefly, isolated cells were recovered and plated at low density in SILAC DMEM-Flex (Invitrogen), lacking arginine (R) and lysine (K) and supplemented with 10% dialyzed fetal bovine serum (dFBS) (Invitrogen), 4.5 g/L glucose, 2mM L-glutamine, 100 units/mL penicillin and 100 μg/mL streptomycin (Merck, Darmstadt, Germany). In the case of light media, standard _L_-lysine and _L_-arginine were used, while isotope-labeled _D_-lysine (D_4_) and _C_-arginine (^13^C_6_), and isotope-labeled _L_-lysine (^13^C_6_,^15^N_2_) and _L_-arginine (^13^C_6_,^15^N_4_) were used for medium and heavy conditions, respectively. For the initial cell expansion, 5 × 10^4^ AC and SF from each patient were seeded in three T-25 and T-75 cell culture flasks, respectively (one flask per condition: light, medium and heavy, for each cell type). At confluence, cells were recovered from each culture flask by trypsinization and plated into 6-well transwells (Corning Inc., Corning, NY, USA): AC were seeded into 6-well plates (5 × 10^4^ AC per well) and SF were seeded in 6-well inserts (1.5 × 10^4^ SF per insert), separately.

### 4.4. Co-Cultures and Treatments

When 80% of confluence was reached, corresponding to 100% of the labeling as previously described [154], the FBS-containing medium was removed and cells were washed thoroughly to remove abundant serum proteins. The inserts with adherent SF were placed onto the corresponding 6-well plates with AC and incubated in serum-free medium with and without 10 nM 45 kDa Fn-fs (Merck) in the presence and absence of 10 nM VIP (Bachem, Bubendorf, CH, Switzerland) for 48 h. Experiments were performed in replicate: one replicate of forward (Fn-fs medium/Fn-fs+VIP heavy) and one replicate of reverse labeling (Fn-fs heavy/Fn-fs+VIP medium) (Figure 6).

### 4.5. Collection and Preparation of Conditioned Media, One-Dimensional Gel Electrophoresis and In-Gel Digestion of Proteins

Secretomes from each condition (light, medium and heavy) were mixed 1:1 and concentrated, using Amicon^®^ Ultra 10K centrifugal filter devices (Merck). About 20 µg were separated on a 10% SDS-PAGE gel. The gel was stained with Coomassie blue and the resulting lanes were cut into 7 slices (Figure 7), and subjected to in-gel digestion as previously described [154]. Briefly, in-gel reduction was done for 45 min at 56 °C using 10 mM DTT in 25 mM ammonium bicarbonate followed by in-gel alkylation, using 55 mM iodoacetamide in 25 mM ammonium bicarbonate for 30 min in the dark. Digestion was performed overnight with 12.5 ng/L Sequencing grade Modified Trypsin (Promega, Madison, WI, USA) at 37 °C. The extracted peptide mixtures were desalted and concentrated via NuTips (Glygen Corp., Columbia, MD, USA) as previously described [154].

### 4.6. NanoLC-MALDI-TOF/TOF Analysis

Peptide fractions were resolved, using RP-nLC in a Tempo nanoLC system (Eksigent, Dublin, CA, USA). The peptide mixture (5 µL) was injected on a C18 precolumn (Michrom, 0.5 × 2 mm) coupled to a reversed-phase column (Integrafit C18, Proteopep II, 75 µm id, 10.2 cm, 5 µm, 300 Å pore size; New Objective, Woburn, MA, USA). Peptides were then eluted in a linear gradient of 5–50% ACN (45 min gradient, flow rate = 350 nL/min), mixed with α-cyano-4-hydroxycinnamic acid matrix (4 mg/mL, flow rate = 1.2 µL/min) and deposited onto a MALDI plate (Sun Collect; Sunchrom, Friedrichsdorf, Germany). Chromatograms corresponding to each gel section were composed of 180 spots. Data acquisition was carried out using a 4800 MALDI-TOF/TOF instrument (Sciex, Foster City, CA, USA), employing the 4000 Series ExplorerTM software version 3.7 (Sciex). MS full-scan spectra from 800 to 4000 m/z were acquired for each peptide-containing LC spots, using 1500 laser shots and a laser intensity of 3800 kV. After screening of all the spots in MS-positive reflector mode, the fragmentation of automatically selected precursors was performed at a collision energy of 1 kV with a collision-induced dissociation (CID) gas (air). Up to 12 of the most intense ion signals per spot position with signal/noise ratios (S/N) above 80 were selected as precursors for tandem mass spectrometry (MS/MS) analysis, excluding common trypsin autolysis peaks and matrix peaks. The number of shots was 1800 for MS/MS, and the laser intensity was set to 4700 kV. A second MS/MS was performed, excluding the precursors selected in the previous MS/MS run. Precursors with S/N > 50 were selected to identify proteins that were not identified in the first MS/MS analysis.

### 4.7. Proteomics Data Analysis

Protein identification was carried out by ProteinPilot 4.5 (Sciex) by searching in the UniProtKB/Swiss-Prot database 2017_02 (http://www.expasy.ch/sprot; 553,655 sequences). Alternatively, hits were contrasted with UniProt_all database for the estimation of contaminating events. Search parameters included SILAC labeling, IAA alkylation, in-gel trypsin digestion and urea denaturalization as specific factors. 

Proteomic data analysis was performed on four different OA biological replicates and two technical replicates by isotope switching. Protein identification and quantification were carried out using ProteinPilot™ software v.4.5 (Sciex). Each MS/MS spectrum was searched in the Uniprot/Swissprot database (UniProt 2017_02 release version containing 553,655, with taxonomy restriction_Homo sapiens), using the Paragon Algorithm. The following ProteinPilot search parameters were used: sample type set as SILAC (Lys +6, Arg +10), oxidation of methionine residue as variable modification, iodoacetamide alkylation of cysteine residue as a fixed modification, in-gel trypsin digestion and urea denaturalization as specific factors and a maximum of one missed cleavage allowed for trypsin. Proteins identified with ≥2 distinct peptides with 95% confidence and a ProtScore ≥1.3 were considered for relative quantification. Proteins identified with one single peptide were manually inspected. Quantified proteins were considered significant when protein ratios ≥ 1.2 or ≤0.83, and *p*-value < 0.05. Data were also normalized for loading error by bias correction. Searches against a concatenated database containing both forward and reversed sequences allowed the false discovery rate to be kept at 1%. Common contaminants, such as albumin, were excluded from the analysis. The results obtained from ProteinPilot were exported to Microsoft Excel for further analyses. Gene ontology enrichment analysis on the set of detected and modulated proteins was performed, using PANTHER and STRING databases, respectively.

### 4.8. ELISA and Multiplex Analysis

Isolated AC and SF were plated into 12-well transwells (Corning): AC were seeded into 12-well plates (2 × 10^5^ AC per well) and SF into 12-well inserts (4 × 10^4^ SF per insert). When 80% confluence was reached, the inserts with adherent SF were placed onto the corresponding 12-well plates with AC and incubated in a serum-free medium with or without 10 nM 45 kDa Fn-fs (Merck) in the presence or absence of 10 nM VIP (Bachem) for 48 h. Secretomes were collected from the cultures and selected proteins were measured, using commercial ELISA kits for CHI3L1, C1r, DCN (Merck), and CTSB (Vitro S.A., Madrid, Spain), and a 3-Plex Multiplex for human MMP-2, PTX-3 and C3 (Merck), following manufacturer’s instructions (Figure 8b). AC and SF were also plated separately into 12-well plates and inserts, respectively, for comparisons between the secretome of each cell type and the co-culture under basal conditions (Figure 8a). In parallel, isolated cultures of SF and AC were also performed in 12-well plates (2 × 10^5^ cells per well) with the same treatments described above, for analysis by ELISA and Multiplex (Figure 8c).

A schematic representation of the whole experimental design is shown in Figure 9.

### 4.9. Statistical Analysis

In the proteomic analysis, normalization tools and the statistical package from Protein Pilot software were employed. In the ELISAS and Multiplex, statistical analysis was performed, using GraphPad Prism software version 8 (GraphPad Inc, San Diego, CA, USA). Data were subjected to a normality test (Kolmogórov–Smirnov test) and equal variance test (F-test). Differences were assessed using Student’s two-tailed t test or Mann–Whitney test for two group comparisons. For comparisons between more than two groups, One-way analysis of variance (ANOVA), Brown–Forsythe or Kruskal–Wallis tests were performed, with Turkey, Games–Howell or Dunn’s multiple comparisons post hoc tests, respectively. Results are presented as mean ± SEM. The *p*-values < 0.05 were considered statistically significant. 

## 5. Conclusions

The present study shows an analysis of the molecules released by SF and AC in the OA knee joint, corroborating the importance of the presence of both cell populations in the initiation and progression of the disease. In addition, our results support the anti-inflammatory and anti-catabolic properties of VIP in rheumatic diseases. VIP-modulated proteins reported in this study would provide potential new targets for OA treatment.

## Figures and Tables

**Figure 1 ijms-22-06441-f001:**
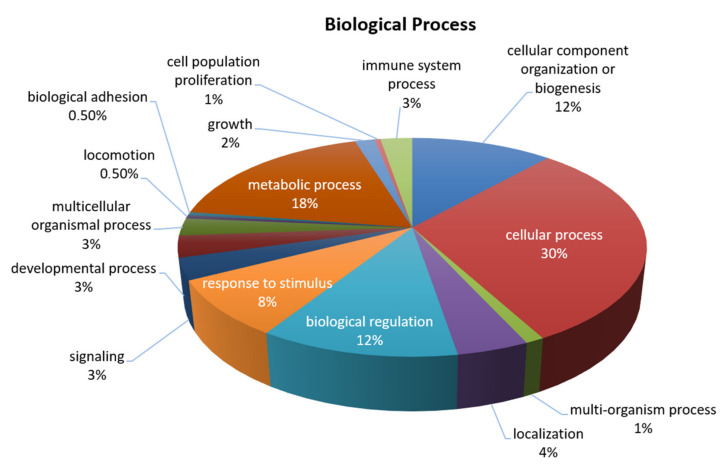
General biological processes involving the proteins detected in the OA SF-AC co-cultures secretome identified by SILAC. Protein classification was performed, using the PANTHER (protein analysis through evolutionary relationships) database.

**Figure 2 ijms-22-06441-f002:**
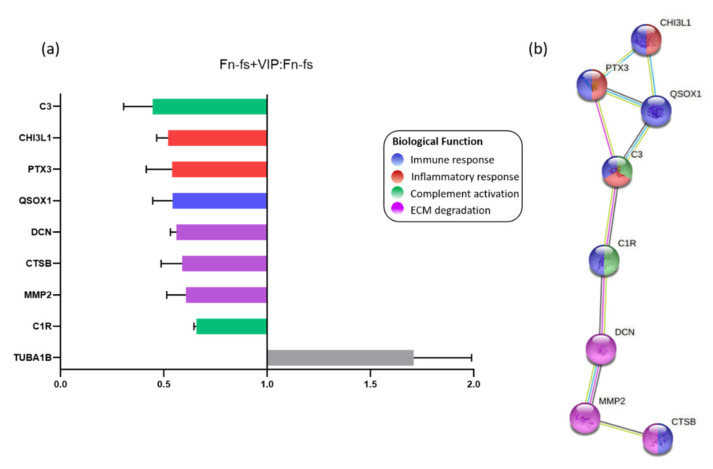
VIP-modulated proteins in the SF-AC co-cultures secretome. (**a**) Average SILAC ratios (*n* = 4) that represent the relative protein abundance in Fn-fs+VIP versus Fn-fs treated SF-AC co-cultures at the same time point (48 h). Ratios are presented as the media of the replicates forward and reverse for each patient. (**b**) Functional protein association networks and biological processes where the proteins downregulated by VIP are involved, according to the STRING database.

**Figure 3 ijms-22-06441-f003:**
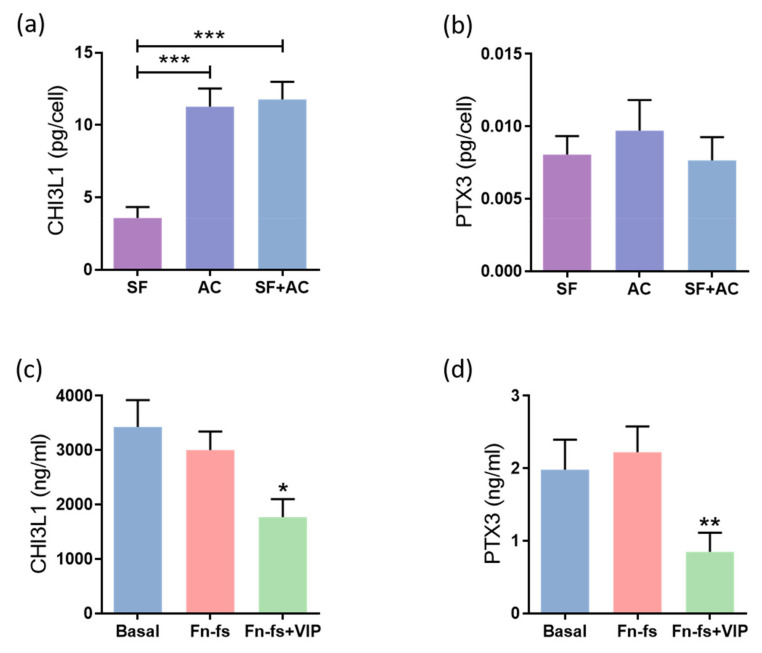
VIP-modulated inflammatory proteins. Secreted levels of (**a**,**c**) CHI3L1 and (**b**,**d**) PTX3 were determined by ELISA and Multiplex, respectively, in the secretomes from SF and AC alone and in co-culture (*n* = 6). Data are presented as mean ± SEM of triplicate determinations. (**a**,**b**) Constitutive protein expression of (**a**) CHI3L1 and (**b**) PTX3 in the secretomes from SF and AC alone and in co-culture at 48 h. Results are presented as pg corrected by the number of cells for each condition. *** *p* < 0.001. (**c**,**d**) Protein expression of (**c**) CHI3L1 and (**d**) PTX3 in the SF-AC co-cultures secretomes at 48 h of treatment with and without 10 nM 45 kDa Fn-fs in the presence and absence of 10 nM VIP. * *p* < 0.05, ** *p* < 0.01 Fn-fs+VIP vs. Fn-fs.

**Figure 4 ijms-22-06441-f004:**
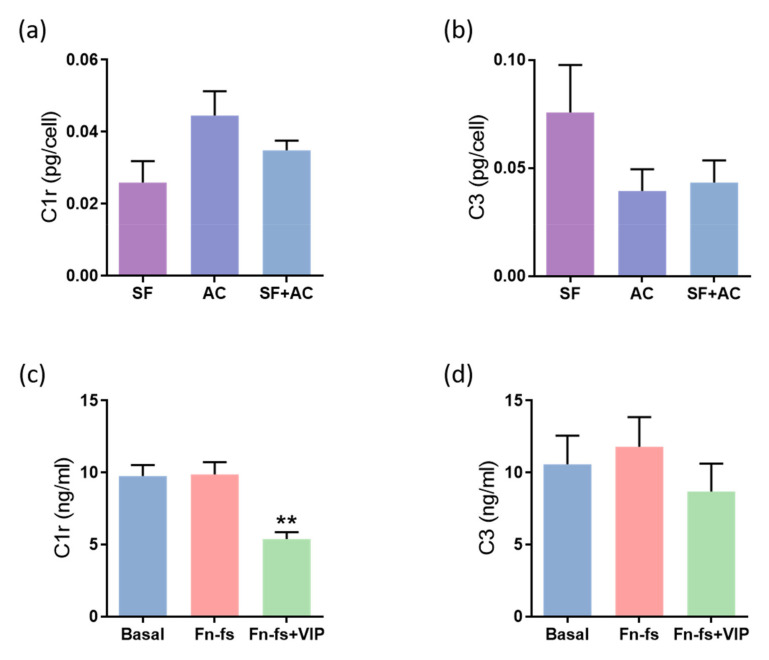
VIP-modulated complement system proteins. Secreted levels of (**a**,**c**) C1r and (**b**,**d**) C3 were determined by ELISA and Multiplex, respectively, in the secretomes from SF and AC alone and in co-culture (*n* = 6). Data are presented as mean ± SEM of triplicate determinations. (**a**,**b**) Constitutive protein expression of (**a**) C1r and (**b**) C3 PTX3 in the secretomes from SF and AC alone and in co-culture at 48 h. Results are presented as pg corrected by the number of cells for each condition. (**c**,**d**) Protein expression of (**c**) C1r and (**d**) C3 in the SF-AC co-cultures secretomes at 48 h of treatment with and without 10 nM 45 kDa Fn-fs in the presence and absence of 10 nM VIP. ** *p* < 0.01 Fn-fs+VIP vs. Fn-fs.

**Figure 5 ijms-22-06441-f005:**
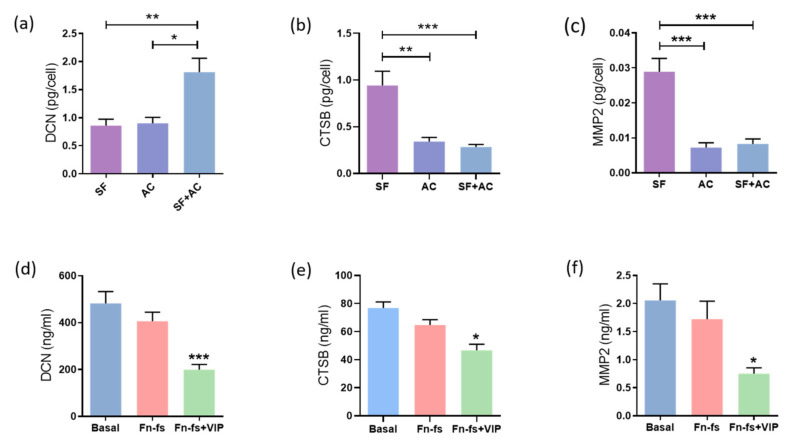
VIP-modulated ECM degradation proteins. Secreted levels of (**a**,**d**) DCN, (**b**,**e**) CTSB, and (**c**,**f**) MMP2 were determined by ELISA and Multiplex, respectively, in the secretomes from SF and AC alone and in co-culture (*n* = 6). Data are presented as mean ± SEM of triplicate determinations. (**a**–**c**) Constitutive protein expression of (**a**) DCN, (**b**) CTSB, and (**c**) MMP2 in the secretomes from SF and AC alone and in co-culture at 48 h. * *p* < 0.05, ** *p* < 0.01, *** *p* < 0.001. Results are presented as pg corrected by the number of cells for each condition. (**d**–**f**) Protein expression of (**d**) DCN, (**e**) DCN, and (**f**) MMP2 in the SF-AC co-cultures secretomes at 48 h of treatment with and without 10 nM 45 kDa Fn-fs in the presence and absence of 10 nM VIP. * *p* < 0.05, *** *p* < 0.001 Fn-fs+VIP vs. Fn-fs.

**Figure 6 ijms-22-06441-f006:**
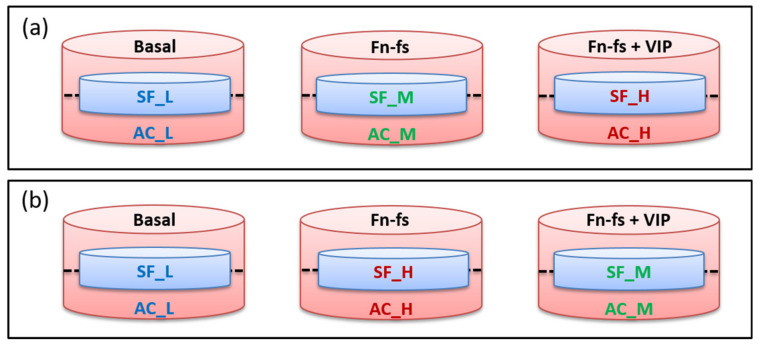
Experimental design of SF-AC co-cultures for SILAC analysis. (**a**) Forward replicate (Fn-fs medium/Fn-fs+VIP heavy). (**b**) Reverse replicate (Fn-fs heavy/Fn-fs+VIP medium). SF, synovial fibroblasts; AC, articular chondrocytes; L, light; M, medium; H, heavy; Fn-fs, fibronectin fragments; VIP, vasoactive intestinal peptide.

**Figure 7 ijms-22-06441-f007:**
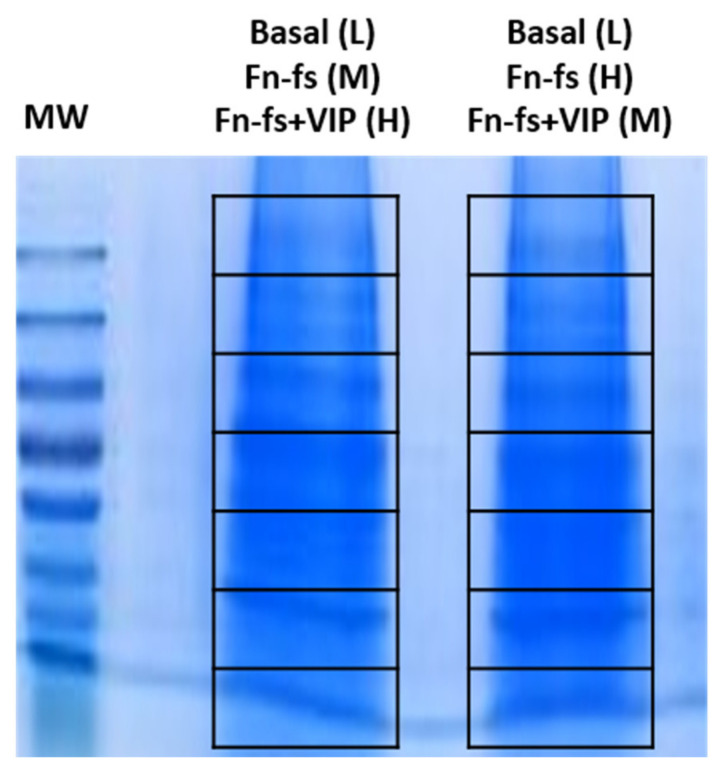
One-dimensional gel electrophoresis separation of SF-AC co-cultures secretome proteins. In each lane, 20 μg of mixed samples (L + M + H) were resolved. Then, the whole gel column was sliced into 7 sections and each lane was subjected to nanoLC-MS/MS analysis. MW, molecular weight markers; L, light; M, medium; H, heavy; Fn-fs, fibronectin fragments; VIP, vasoactive intestinal peptide.

**Figure 8 ijms-22-06441-f008:**
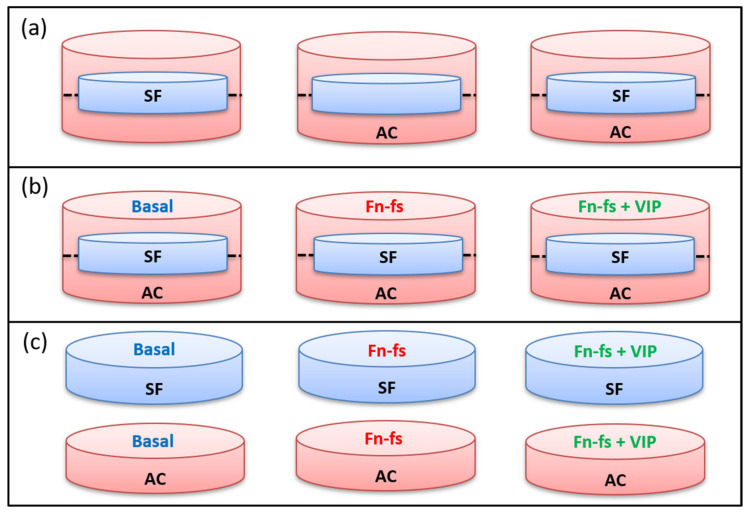
Experimental design of SF and AC co-cultures and isolated cultures for ELISA and Multiplex analysis. (**a**) Basal conditions in the SF-AC transwells. (**b**) Treatments of SF-AC co-cultures. (**c**) Treatments of SF and AC isolated cultures. SF, synovial fibroblasts; AC, articular chondrocytes; Fn-fs, fibronectin fragments; VIP, vasoactive intestinal peptide.

**Figure 9 ijms-22-06441-f009:**
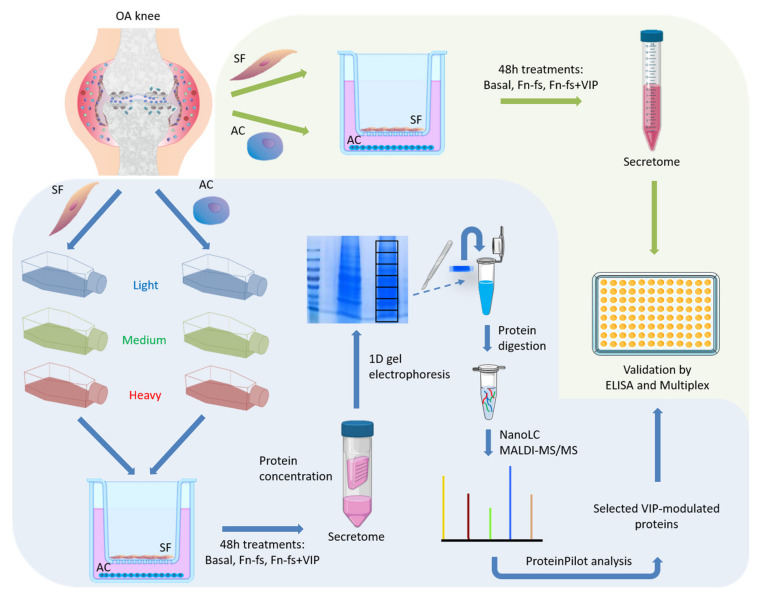
Schematic representation of the experimental design.

**Table 1 ijms-22-06441-t001:** Proteins identified by SILAC analysis in the secretome of OA SF-AC co-cultures.

Acc No ^a^	Protein Symbol	Protein Name	Biological Function
P01023	A2MG	Alpha-2-macroglobulin	Immune response
P63261	ACTG	Actin, cytoplasmic 2	Cytoskeletal protein
P12814	ACTN1	Alpha-actinin-1	Cytoskeletal protein
P04075	ALDOA	Fructose-bisphosphate aldolase A	Cellular metabolic process
P15144	AMPN	Aminopeptidase N	Immune response
P01008	ANT3	Antithrombin-III	Angiogenesis
P27695	APEX1	DNA-(apurinic or apyrimidinic site) lyase	Oxidative stress
Q9P1U1	ARP3B	Actin-related protein 3B	Cytoskeletal organization
P61769	B2MG	Beta-2-microglobulin	Immune response
Q15582	BGH3	Transforming growth factor-beta-induced protein ig-h3	TGF-beta signaling
P00736	C1R	Complement C1r subcomponent	Complement activation
P09871	C1S	Complement C1s subcomponent	Complement activation
P40121	CAPG	Macrophage-capping protein	Macrophage function
P07858	CATB	Cathepsin B	ECM degradation
P07339	CATD	Cathepsin D	ECM degradation
P07711	CATL1	Cathepsin L1	ECM degradation
P16070	CD44	CD44 antigen	ECM degradation/Inflammatory response
P00751	CFAB	Complement factor B	Complement activation
P08603	CFAH	Complement factor H	Complement activation
P36222	CH3L1	Chitinase-3-like protein 1	Inflammatory response
P10909	CLUS	Clusterin	Immune response/Complement activation
Q8N137	CNTRB	Centrobin	Centriole duplication
P08123	CO1A2	Collagen alpha-2(I) chain	ECM component
P01024	CO3	Complement C3	Complement activation
P12109	CO6A1	Collagen alpha-1(VI) chain	ECM component
P12111	CO6A3	Collagen alpha-3(VI) chain	ECM component
Q99715	COCA1	Collagen alpha-1(XII) chain	ECM component
P49747	COMP	Cartilage oligomeric matrix protein	ECM component
Q14019	COTL1	Coactin-like protein	Cytoskeletal protein binding
P02511	CRYAB	Alpha-crystallin B chain	Cytoskeletal protein binding
O94985	CSTN1	Calsyntenin-1	Cytoskeletal protein binding
P13639	EF2	Elongation factor 2	Cytoskeletal protein binding
P06733	ENOA	Alpha-enolase	Immune response
Q12805	FBLN3	EGF-containing fibulin-like extracellular matrix protein 1	ECM component/Negative regulator of chondrocyte differentiation
P02751	FINC	Fibronectin	ECM component/ECM degradation
Q06828	FMOD	Fibromodulin	ECM component
P04406	G3P	Glyceraldehyde-3-phphate dehydrogenase	Cellular metabolic process/Immune response
P06744	G6PI	Gluce-6-phphate isomerase	Cellular metabolic process/Immune response
P50395	GDIB	Rab GDP dissociation inhibitor beta	Immune response
P07093	GDN	Glia-derived nexin	ECM component/Serine protease inhibitor
P28161	GSTM2	Glutathione S-transferase Mu 2	Cellular metabolic process/Inflammatory response
P09211	GSTP1	Glutathione S-transferase P	Cellular metabolic process/Inflammatory response
P57053	H2BFS	Histone H2B type F-S	Immune response
P02042	HBD	Hemoglobin subunit delta	Oxygen transport
P11142	HSP7C	Heat shock cognate 71 kDa protein	Immune response
P04792	HSPB1	Heat shock protein beta-1	Immune response
P17936	IBP3	Insulin-like growth factor-binding protein 3	Cell proliferation and differentiation
P22692	IBP4	Insulin-like growth factor-binding protein 4	Cell proliferation and differentiation
P24592	IBP6	Insulin-like growth factor-binding protein 6	Cell proliferation and differentiation
Q16270	IBP7	Insulin-like growth factor-binding protein 7	Cell proliferation and differentiation
P05155	IC1	Plasma protease C1 inhibitor	Complement activation
P0DOX7	IGK	Immunoglobulin kappa light chain	Immune response
P05231	IL6	Interleukin-6	Inflammatory response
P13645	K1C10	Keratin, type I cytoskeletal 10	Cytoskeletal protein
P35527	K1C9	Keratin, type I cytoskeletal 9	Cytoskeletal protein
P04264	K2C1	Keratin, type II cytoskeletal 1	Cytoskeletal protein
P14618	KPYM	Pyruvate kinase PKM	Cellular metabolic process/Immune response
P00338	LDHA	L-lactate dehydrogenase A chain	Cellular metabolic process
P17931	LEG3	Galectin-3	Inflammatory response
P51884	LUM	Lumican	ECM component
P33908	MA1A1	Mannosyl-oligosaccharide 1,2-alpha-mannidase IA	Cellular metabolic process
P40925	MDHC	Malate dehydrogenase, cytoplasmic	Cellular metabolic process
P14174	MIF	Macrophage migration inhibitory factor	Immune response
P03956	MMP1	Interstitial collagenase	ECM degradation
P08253	MMP2	72 kDa type IV collagenase	ECM degradation
P08254	MMP3	Stromelysin-1	ECM degradation
P26038	MOES	Moesin	Cytoskeletal protein binding
P22392	NDKB	Nucleoside diphosphate kinase B	Immune response
Q96TA1	NIBL1	Niban-like protein 1	Apoptosis suppression
P05121	PAI1	Plasminogen activator inhibitor 1	Serine protease inhibitor
Q15113	PCOC1	Procollagen C-endopeptidase enhancer 1	Cellular metabolic process
P30101	PDIA3	Protein disulfide-isomerase A3	Cellular metabolic process
Q15084	PDIA6	Protein disulfide-isomerase A6	Cellular metabolic process
P18669	PGAM1	Phosphoglycerate mutase 1	Cellular metabolic process
P00558	PGK1	Phosphoglycerate kinase 1	Cellular metabolic process
P21810	PGS1	Biglycan	ECM component/ECM degradation
P07585	PGS2	Decorin	ECM component/ECM degradation
O60664	PLIN3	Perilipin-3	Protein transport
P62937	PPIA	Peptidyl-prolyl cis-trans isomerase A	Cellular metabolic process
P23284	PPIB	Peptidyl-prolyl cis-trans isomerase B	Cellular metabolic process
Q06830	PRDX1	Peroxiredoxin-1	Immune response
P30041	PRDX6	Peroxiredoxin-6	Immune response
Q92954	PRG4	Proteoglycan 4	ECM component/Control of synovial growth and adhesion of to the cartilage surface
P07737	PROF1	Profilin-1	Cytoskeletal protein binding
P49721	PSB2	Proteasome subunit beta type-2	Immunoproteasome assembly
Q9UL46	PSME2	Proteasome activator complex subunit 2	Immunoproteasome assembly
P26022	PTX3	Pentraxin-related protein 3	Inflammatory response
O00391	QSOX1	Sulfhydryl oxidase 1	Oxidation-reduction process
P55017	S12A3	Solute carrier family 12 member 3	Inflammatory response
P50454	SERPH	Serpin H1	ECM organization
Q9H299	SH3L3	SH3 domain-binding glutamic acid-rich-like protein 3	Oxidation-reduction process
P04179	SODM	Superoxide dismutase [Mn], mitochondrial	Oxidation-reduction process
P09486	SPRC	SPARC	Cell proliferation and differentiation
P42224	STAT1	Signal transducer and activator of transcription 1-alpha/beta	Immune response
P23381	SYWC	Tryptophan--tRNA ligase, cytoplasmic	Angiogenesis
P68363	TBA1B	Tubulin alpha-1B chain	Cytoskeletal protein
Q13509	TBB3	Tubulin beta-3 chain	Cytoskeletal protein
P07437	TBB5	Tubulin beta chain	Cytoskeletal protein
P24821	TENA	Tenascin	ECM component
P01033	TIMP1	Metalloproteinase inhibitor 1	Metalloprotease inhibitor
P16035	TIMP2	Metalloproteinase inhibitor 2	Metalloprotease inhibitor
P29401	TKT	Transketolase	Oxidation-reduction process
P60174	TPIS	Triosephosphate isomerase	Oxidation-reduction process
P02787	TRFE	Serotransferrin	Iron binding transport protein
P02788	TRFL	Lactotransferrin	Iron binding transport protein
Q8NBS9	TXND5	Thioredoxin domain-containing protein 5	Immune response
P19971	TYPH	Thymidine phosphorylase	Angiogenesis
P22314	UBA1	Ubiquitin-like modifier-activating enzyme 1	Proteasome degradation
O60701	UGDH	UDP-glucose 6-dehydrogenase	Biosynthesis of ECM components
Q6EMK4	VASN	Vasorin	TGF-beta signaling
P08670	VIME	Vimentin	Cytoskeletal protein/Immune response
P18206	VINC	Vinculin	ECM binding/Immune response
P02774	VTDB	Vitamin D-binding protein	Vitamin D transport and storage
P04004	VTNC	Vitronectin	ECM binding/Immune response
Q5GH72	XKR7	XK-related protein 7	XK related family

^a^ Protein accession number according to the SwissProt and TrEMBL databases. SILAC, stable isotope labeling by amino acids in cell culture; OA, osteoarthritis; SF, synovial fibroblasts; AC, articular chondrocytes.

**Table 2 ijms-22-06441-t002:** VIP-modulated proteins in the in the Fn-fs-stimulated secretome of OA SF-AC co-cultures identified by SILAC.

Acc No ^a^	Protein Symbol	Gene Symbol	Protein Name	Ratio ^b^	
				Forward	Reverse
P01024	CO3	C3	Complement C3	0.545	0.346
P36222	CH3L1	CHI3L1	Chitinase-3-like protein 1	0.481	0.559
P26022	PTX3	PTX3	Pentraxin-related protein 3	0.451	0.628
O00391	QSOX1	QSOX1	Sulfhydryl oxidase 1	0.610	0.474
P07585	PGS2	DCN	Decorin	0.582	0.540
P07858	CATB	CTSB	Cathepsin B	0.660	0.516
P08253	MMP2	MMP2	72 kDa type IV collagenase	0.672	0.541
P00736	C1R	C1R	Complement C1r subcomponent	0.666	0.649
P68363	TBA1B	TUBA1B	Tubulin alpha-1B chain	1.907	1.511

^a^ Protein accession number according to the SwissProt and TrEMBL databases. ^b^ SILAC ratios that represent the relative protein abundance in Fn-fs+VIP versus Fn-fs treated SF-AC co-cultures at the same time point (48 h) (*n* = 4). *p*-values and error factors are shown in Appendix A. VIP, vasoactive intestinal peptide; Fn-fs, fibronectin fragments; OA, osteoarthritis; SF, synovial fibroblasts; AC, articular chondrocytes; SILAC, stable isotope labelling by amino acids in cell culture.

## Data Availability

The data presented in this study are available on request from the corresponding author. The data are not publicly available due to the lack of an existing repository.

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
