# Peer review of "Proteomic Analysis of Synovial Fibroblasts and Articular Chondrocytes Co-Cultures Reveals Valuable VIP-Modulated Inflammatory and Degradative Proteins in Osteoarthritis"

_ijms, 2021, doi:10.3390/ijms22126441_

Round 1
Reviewer 1 Report
In this manuscript, authors described VIP may be play a critical role in the pathogenesis of OA. In addition, this manuscript reviews the effects of VIP on inflammatory biomarkers.
1) In Fig. 2b, the author shows the association of VIP-modulated protein, but the association with complement other than ECM and inflammatory response should also be considered.
2) In Fig. 3, 4 and 5, the action of VIP was evaluated in the presence 45KDa Fn-fs and VIP, but the effect of 45KDa Fn-fs alone on inflammatory response, complement and ECM protein was not observed compared to the basesal. The author should explain this in detail. The author should also evaluate the action of VIP alone and the dose dependence of VIP.
3) In Abstract and discussion, the author also mentions the importance of VIP for rheumatoid diseases, which is unclear and overstated from this manuscript.
4) The authors were investigating the effects of VIP on secreted proteins in AC and SF co-cultures. However, VIP appears to act on SF. The author should explain the need for co-culture with OA chondrocytes.
Author Response
"Please see the attachment"

Reviewer 2 Report
Pérez-García et al aimed to explore new biomarkers of osteoarthritis (OA) by analysing set of secreted proteins in articular cartilage and synovial fibroblasts (SF) co-cultured by Stable Isotope Labeling using Amino acids in Cell culture (SILAC) method. They identified 76 proteins exclusively detected in these co-cultures stimulated by fibronectin fragments. Furthermore, they also studied the role of the vasointestinal peptide (VIP) in this secretome. Using ELISA and multiplex analyses they showed that VIP decreases proteins involved in the inflammatory process (CHI3L1, PTX3), complement activation (C1r, C3), and cartilage extracellular matrix degradation (DCN, CTSB and MMP2). All of these are key events in the initiation and progression of OA. Their results support the anti-inflammatory and anti-catabolic properties of VIP in the pathogenesis of OA and provide potential new targets for OA treatment.
The inflammatory process and the role SF play in rheumatoid arthritis is well-established, however the inflammatory component of the degenerative, i.e. the joint wear and tear disease, OA has been only recently suggested. Therefore the aim of the study is of particular interest. The study is also well designed, the experimental procedures are sound, the results are presented in appropriate way and the paper is very well written. I would only have a few minor comments pointed our below:
- The authors user the term ”rheumatic disease” for OA. Since this term is more common for the RA, I would suggest the authors to replace this word with more appropriate one. In my opinion inflammatory would be the best option here.
- The method used in the current study, i.e. Stable isotope labeling using amino acids in cell culture (SILAC) method, is relatively new and very specific, so its abbreviations should be defined at first use and also in the Abstract. This abbreviation is also missing in the list of Abbreviations.
- Could the authors please provide some reference to the American College of Rheumatology (ACR) criteria for OA classification under Materials and Methods.
- There are also typos in 1% l‐glutamine and 5% CO2. Please correct to L-glutamine and CO
- Could the authors please provide the reference for creation of schematic representation as shown in Figure 9.
Author Response
"Please see the attachment"

Reviewer 3 Report
The manuscript entitled “Proteomic analysis of synovial fibroblasts and articular chondrocytes co-cultures reveals valuable VIP-modulated inflammatory and degradative proteins in osteoarthritis” is an interesting study about the effects of VIP on an in vitro model.
However, I have several comments for the authors to improve their study.
In the abstract, the authors reported that they identified 76 proteins exclusively detected in SF-AC co-cultures stimulated by fibronectin fragments (Fn-fs). However, this part is not described in the results section. In the results the authors reported that they identified 115 proteins in the co-cultures and then the authors focused on downregulated proteins in the presence of the inflammatory stimulus and VIP.
The introduction should be improved. In particular, there is no mention about the fact that also the meniscus and the infrapatellar fat pad have a role in OA. Moreover, both tissues are able to secrete inflammatory molecules (doi.org/10.1080/03008207.2018.1470167).
The introduction and also the discussion about VIP should be improved. There are contradictory studies published on VIP effects as explained in the reference number 22 cited by the authors. It seems for example that VIP is involved in OA pain.
In the results, there is no description about the effects of the treatment with Fn-fs of the co-culture. This part should be added. Could the authors provide the list of proteins identified by SILAC in the Fn-fs stimulated co-cultures and the list proteins identified in the Fn-fs stimulated plus VIP co-cultures?
It is not clear to me why the authors presented the Results as pg corrected by the number of cells for each condition only for SF, AC and SF+AC, while the authors reported ng/mL for co-cultures alone and treated with Fn-fs and Fn-fs+VIP (figures 3,4 and 5). Why did the authors use two different methods?
It is not clear to me why the authors focused only on down-regulated proteins. Is it possible to add a list of up-regulated proteins?
Lines 342-345: the authors reported that “despite no significant effects have been detected in the proteins analysed in this study with the Fn-fs treatment alone, this stimulus is necessary for VIP to exert its effects”. This part needs to be better elucidated. Why is it necessary to stimulate the cells to obtain a VIP effect? Why did the authors decide to use Fn-fs as inflammatory stimulus and not for example IL-1b or TNF-alpha?
Line 380-381: “Osteoarthritic cartilage and synovial tissue were obtained from the knee of ten patients 380 (five women and five men) aged between 50 and 92, undergoing leg amputation.” Patients with end-stage OA undergo to total joint replacement. It is not clear to me why these OA patients underwent leg amputation. The body mass index of the patients should be reported.
Line 396: “CO2” should be corrected.
Lines 492-493:” Isolated AC and SF were plated into 6-well transwells (Corning): AC were seeded into 12-492 well plates (2 × 105 AC per well) and SF into 12-well inserts (4 × 104 SF per insert).” It is not clear if cells were plated into 6 wells or 12 wells.
Author Response
"Please see the attachment"

Round 2
Reviewer 1 Report
Thank you for your quick response. The manuscript has been properly revised. There are no more problems.
Reviewer 3 Report
No additional comments.